# Cryopreservation impairs 3-D migration and cytotoxicity of natural killer cells

Christoph Mark[1,6], Tina Czerwinski[1,6], Susanne Roessner[2,3,4], Astrid Mainka[1], Franziska Hörsch [1], Lucas Heublein[1], Alexander Winterl [1], Sebastian Sanokowski [1], Sebastian Richter [1], Nina Bauer[1], Thomas E. Angelini[5], Gerold Schuler[2,3,4], Ben Fabry [1✉] & Caroline J. Voskens [2,3,4]

Natural killer (NK) cells are important effector cells in the immune response to cancer. Clinical trials on adoptively transferred NK cells in patients with solid tumors, however, have thus far been unsuccessful. As NK cells need to pass stringent safety evaluation tests before clinical use, the cells are cryopreserved to bridge the necessary evaluation time. Standard degranulation and chromium release cytotoxicity assays confirm the ability of cryopreserved NK cells to kill target cells. Here, we report that tumor cells embedded in a 3-dimensional collagen gel, however, are killed by cryopreserved NK cells at a 5.6-fold lower rate compared to fresh NK cells. This difference is mainly caused by a 6-fold decrease in the fraction of motile NK cells after cryopreservation. These findings may explain the persistent failure of NK cell therapy in patients with solid tumors and highlight the crucial role of a 3-D environment for testing NK cell function.

[1] Friedrich-Alexander University Erlangen-Nürnberg, Department of Physics, Erlangen, Germany. [2] Friedrich-Alexander University Erlangen-Nürnberg and University Hospital Erlangen, Department of Dermatology, Erlangen, Germany. [3] Comprehensive Cancer Center Erlangen-European Metropolitan Area of Nürnberg (CCC ER-EMN), Erlangen, Germany. [4] Deutsches Zentrum für Immuntherapie (DZI), Erlangen, Germany. [5] University of Florida, Department of Mechanical and Aerospace Engineering, Gainesville, FL, USA. [6] These authors contributed equally: Christoph Mark, Tina Czerwinski. ✉email: ben.fabry@fau.de

Natural killer (NK) cells are important effector cells in the early innate immune response to various pathogens, including cancer. For the elimination of target cells, NK cells form a temporary immune synapse with the target cell and secrete granules containing cell-toxic granzymes and perforin. To reach target cells outside the blood stream, NK cells can extravasate and migrate through the connective tissue of numerous organs[1]. Adoptive transfer of human NK cells in mice has been shown to suppress the development of primary tumors and metastases[2–4], and clinical studies have shown encouraging results in patients with hematological malignancies[5,6]. Studies in patients with solid tumors, however, have thus far failed to demonstrate antitumor responses[7–11], and although the transferred NK cells remain viable in the peripheral circulation, they seem to lose their cytotoxic function in vivo[10].

The cytotoxicity of NK cells is typically evaluated in CD107a degranulation and chromium-release assays[12–14]. The degranulation assay detects CD107a proteins from cytolytic granule membranes that are transported to the surface of NK cells upon the formation of an immune synapse. The chromium-release assay measures the actual number of tumor cells that are lysed by NK cells. In both assays, the NK cells are in close contact with the tumor cells and do not need to migrate far to reach tumor cells. In vivo, however, the ability to infiltrate a three-dimensional (3-D) environment is crucial for NK cells to reach the tumor cells.

Clinical application of NK cells demands stringent evaluation regarding sterility, purity, and function. This requires freezing and thawing of the cells to bridge the necessary time for passing predefined lot-release criteria. Therefore, clinical studies mainly rely on the use of cryopreserved cells. Reports on the effect of cryopreservation on the cytotoxicity of ex vivo-expanded NK cells are mixed, ranging from no significant effects[15] to a more than twofold reduction in cytotoxicity[16].

In this work, we confirm that NK cells retain their ability to induce target cell death in a degranulation assay, but we find a decrease of cytotoxic function in a chromium-release assay after cryopreservation, following a good manufacturing practice conform protocol, which was developed and approved for the cryopreservation of T cells intended for clinical use[17,18]. To investigate the origin of this decreased cytotoxicity, we perform time-lapse imaging, and tracking of NK cells and target cells embedded in 3-D collagen gels that serve as a model system for the extracellular matrix of connective tissue. We find that the fraction of motile NK cells in 3-D collagen gels is decreased by sixfold after cryopreservation, while the small remaining population of motile cells retains its cytotoxic function in a 3-D environment.

## Results

**NK cells are viable and active after cryopreservation.** We confirm by live–dead staining and flow cytometry that cell viability is not affected by the process of freezing and thawing (Fig. 1a). The 14-day expansion protocol of NK cells from peripheral blood mononuclear cell (PBMC) significantly increases the fraction of NK cells (identified as CD56+ CD3−) by a factor of 4, which remains unchanged after cryopreservation (Fig. 1b). However, cryopreservation results in a significant decrease of the CD16+ subpopulation of NK cells (Fig. 1b). This subpopulation is also referred to as activated or cytotoxic NK cells, as the CD16 surface receptor is a known mediator of NK cell cytotoxicity.

To assess whether the relative decrease in the CD16+ subpopulation of NK cells affects the potential antitumor activity of the whole expanded NK cell population, we perform a CD107a degranulation assay. This assay tests the ability of NK cells to fuse lytic granules with their membrane, which is an essential step to

induce target cell death. We find no significant difference in the expression of CD107a between fresh and cryopreserved NK cells, indicating that cryopreserved NK cells retain the ability to induce target cell death (Fig. 1c).

**Cryopreservation reduces number of motile NK cells.** Next, we evaluate the target cell death directly in a coculture of NK cells and K562 leukemia cells across a range of NK-to-target cell ratios, using a chromium-release cytotoxicity assay. After 4 h of incubation time, we find a statistically significant ($p < 0.05$; Spearman's rank-order correlation) decrease in target cell death after cryopreservation compared to fresh NK cells (Fig. 1d). This decrease in cytotoxicity may result from two different mechanisms: first, cryopreservation may affect NK cell activation and induce subsequent cleavage of CD16 by the activation of matrix metalloproteinases[19], as indicated by the reduced number of CD16+ cells. Second, cryopreservation may reduce NK cell motility such that they cannot come in contact with target cells beyond their immediate neighbors.

In support of the second mechanism, we find that the decrease in cytotoxicity after cryopreservation is more pronounced for smaller NK-to-target cell ratios (for the same number of K562 cells). Assuming that only a fraction of NK cells mediates the majority of target cell deaths[20], those NK cells would need to kill a larger number of K562 cells to retain the same overall cytotoxicity in cell populations that contain fewer NK cells. At the

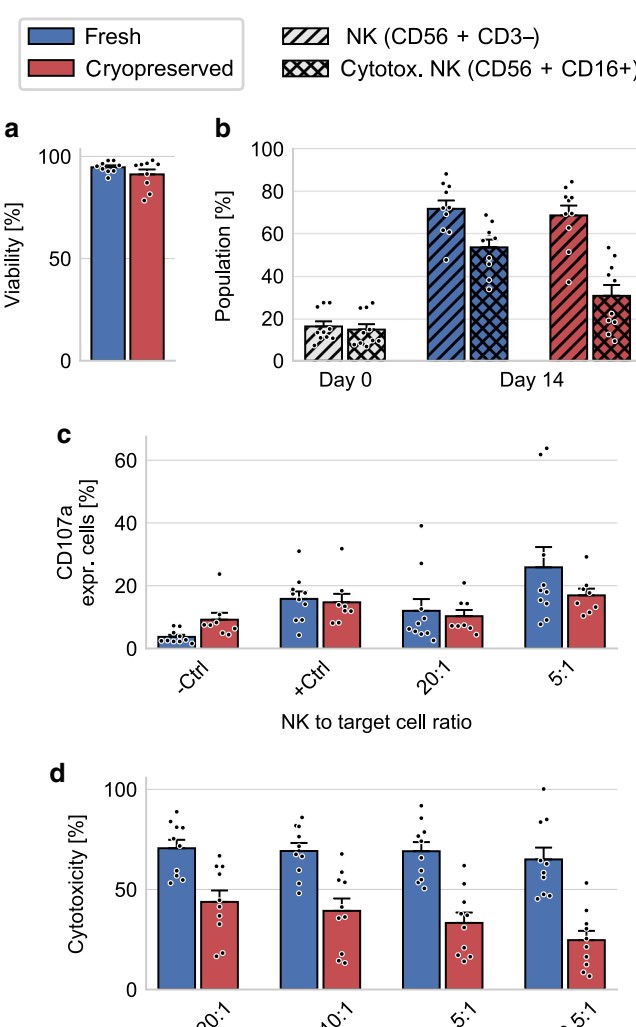

**Fig. 1 NK cell expansion and NK cell function in standard assays. a** NK cell viability of fresh (blue) and cryopreserved (red) NK cells as measured by flow cytometry ($n = 9$; $p = 0.36$ by two-sided nonparametric Wilcoxon signed-rank test for paired data). **b** Fraction of NK cells (single-hatched) and cytotoxic NK cells (double-hatched) within PBMCs before expansion (Day 0; white bars), and after expansion (Day 14) for fresh (blue) and cryopreserved (red) samples ($n = 10$). The increase in the fraction of NK cells and cytotoxic NK cell after expansion is significant ($p = 0.002$ for both conditions by two-sided nonparametric Wilcoxon signed-rank test for paired data). Cryopreservation does not decrease the fraction of NK cells ($p = 0.49$), but decreases the fraction of cytotoxic NK cells ($p = 0.004$ by two-sided nonparametric Wilcoxon signed-rank test for paired data). **c** Percentage of CD107a-expressing NK cells for different NK to target cell ratios, as measured in a degranulation assay ($n = 8$). Control measurements are conducted in the absence of target cells. PMA/Iono is added for positive control experiments. Differences in the degranulation between fresh and cryopreserved NK cells do not reach statistical significance ($-$Ctrl: $p = 0.055$, $+$Ctrl: $p = 0.74$, 20:1: $p = 0.74$, 5:1: $p = 0.38$; two-sided nonparametric Wilcoxon signed-rank test for paired data). **d** NK cell cytotoxicity as measured in a chromium-release assay for different NK to target cell ratios ($n = 10$). Differences between fresh to cryopreserved NK cells reach statistical significance for all conditions (20:1: $p = 0.014$, 10:1: $p = 0.006$, 5:1: $p = 0.004$, 2.5:1: $p = 0.002$; Spearman's rank-order correlation). For fresh NK cells, there is no significant correlation between NK-to-target cell ratio and cytotoxicity (correlation coefficient $\rho = 0.14$, $p = 0.38$; Spearman's rank-order correlation), but for cryopreserved NK cells, cytotoxicity decreases for lower NK-to-target cell ratios (correlation coefficient $\rho = 0.41$, $p = 0.009$; Spearman's rank-order correlation). Error bars denote 1 sem. Source data are provided as a Source data file.

same time, those NK cells would need to migrate over larger distances to reach target cells. For fresh NK cells, by contrast, we find no statistically significant correlation between cytotoxicity and NK-to-target ratio (Fig. 1d), indicating that fresh NK cells can compensate for a decreasing NK-to-target cell ratio, but cryopreserved NK cells cannot. By this rationale, it would be expected that at earlier time points after mixing NK cells and target cells in a cell pellet, serial killing and migration are less important, and the chromium-release cytotoxicity of fresh and cryopreserved cells equalizes cytotoxicity similar to the low NK-to-target ratios. We find this to be true for up to 1 h after cell mixing (Supplementary Fig. 1).

Since the chromium-release assay only reports the decreased cytotoxicity, but cannot measure NK cell motility directly, we perform additional assays for which NK cells are embedded in 3-D reconstituted collagen gels (Fig. 2). To assess the fraction of NK cells that are motile in this 3-D environment, we perform z-scans through a 1 mm thick gel every 30 s, and track NK cell positions by their appearance in minimum intensity projections (Fig. 2a–c, see "Methods" section). We find that NK cell motility after cryopreservation in a 3-D environment is dramatically impaired after cryopreservation. Specifically, we find that 29.2% of fresh NK cells are motile, in line with reported findings for 2-D migration[21], while only 4.9% of cryopreserved NK cells are motile, corresponding to a sixfold decrease in motility (Fig. 3a and Supplementary Fig. 2a).

To more closely mimic the situation in the human body after adoptive transfer of expanded NK cells, we monitor the migration behavior of fresh and cryopreserved NK cells from three subjects over 48 h in the absence of IL-2. The motile fraction of cryopreserved NK cells increases from 2 (6 h after thawing, 4 h after embedding in collagen) to 7% (after 48 h), whereas the motile fraction of fresh cells decreases from 36 (4 h after

embedding in collagen) to 20% (after 48 h; Supplementary Fig. 3a). These data demonstrate that the fraction of motile cryopreserved NK cells does not fully recover even after 48 h, and remains approximately threefold lower compared to fresh cells.

To explore if the negative effect of cryopreservation on 3-D cell migration is specific for the expansion protocol used in this study, we measure the migration of the NK cell line NK92, instead of NK cells expanded from primary PBMCs. NK92 cells migrate in collagen at a similar speed, but slightly less persistent compared to expanded NK cells. Importantly, we find that the motile fraction of NK92 cells decreases from 31 to 6% after cryopreservation, similar to our findings for expanded NK cells (Supplementary Fig. 4a).

Moreover, to explore if the negative effect of cryopreservation on 3-D migration is specific for a 3-D collagen network, we suspend fresh and cryopreserved NK cells in carbomer, a hydrogel-forming polymer based on acrylic acid. Carbomer forms ~10 μm colloidal particles that are jammed to a viscoelastic hydrogel (yield stress 10 Pa) reminiscent of a cell pellet. In contrast to collagen, carbomer does not support the adhesion of cells to the matrix and only allows the cells to migrate in an ameboid mode. Despite the pronounced mechanical and structural differences between both hydrogels, the motile fraction, migration speed, and persistence of fresh NK cells is only slightly lower in carbomer compared to collagen (Supplementary Fig. 5). Importantly, the motile fraction after cryopreservation is strongly reduced in carbomer, to a similar degree as in collagen, demonstrating that the negative effect of cryopreservation on 3-D migration is of a more general nature. This finding is also consistent with the above argument that the reduced cytotoxicity after cryopreservation, as measured in a chromium-release assay is largely attributable to a reduced motile fraction of the NK cells in a cell pellet.

As chromium-release assays and motility assays are performed pair wise, we can quantify the relationship between cytotoxicity and motile cell fraction in 3-D collagen gels for individual samples (specified by donor and expansion). We find that cytotoxicity increases with motile fraction according to a power-law relationship with exponents ranging from 0.2 to 0.4. This relationship reaches statistical significance, with a coefficient of determination of $R^2 \geq 0.42$ for all NK-to-target cell ratios between 20:1 and 2.5:1 (Fig. 3b). This finding adds further support to the notion that NK cells are able to kill target cells beyond their immediate neighbors in a cell pellet (e.g., in a chromium-release assay), for which the ability to migrate is of advantage.

**Motile NK cell fraction retains killing efficiency**. We next investigate whether the small remaining population of motile NK cells after cryopreservation retains full cell function. First, we characterize cell speed and directional persistence of fresh and cryopreserved motile NK cells within 3-D collagen gels, and find no significant difference (Fig. 4a, b, and Supplementary Figs. 2b, c, 3b, c and 5b, c). Thus, NK cells that remain motile after cryopreservation retain their normal exploration behavior within tissue.

Second, we evaluate the cytotoxic function of the motile NK cell fraction in 3-D collagen gels by identifying individual K562 target cell killing events in time-lapse image series (Fig. 2d, e, Supplementary Movies 1 and 3, and see "Methods" section). We confirm by microscopy inspection that each killed K562 cell was attacked by an NK cell during the last 60 min before cell death. The number of live target cells decreases exponentially over time, which can be modeled by a first-order differential equation (see Eq. (1) in "Methods" section). The characteristic time constant of the exponential decay, which reports the average killing rate and hence the cytotoxicity of the motile NK cells, can be estimated

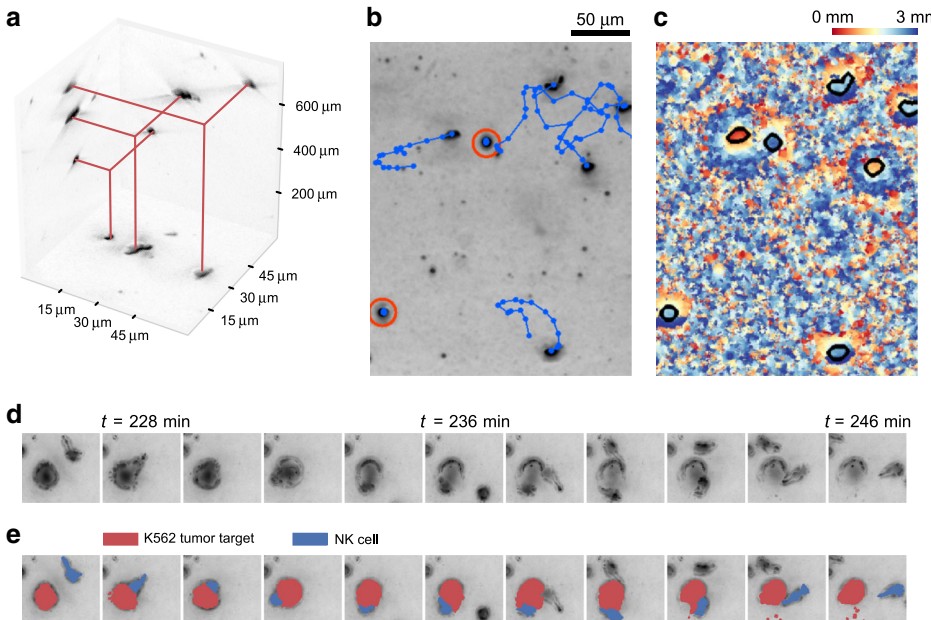

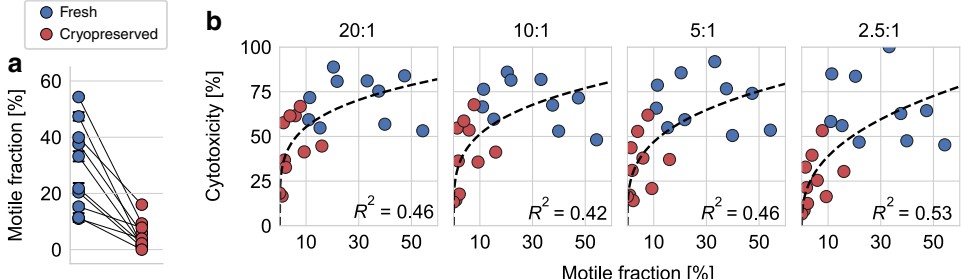

**Fig. 2 NK cell motility assay and cytotoxicity assay in a 3-D collagen gel. a** Minimum intensity projections along the *x/y/z*-axes of a bright-field image stack of NK cells migrating through a 3-D collagen gel. In the projections, the NK cells appear as dark spots. **b** Exemplary minimum intensity projection along the *z*-axis, taken from one out of ten similar, independent experiments. Cell trajectories over 30 min are indicated in blue, nonmotile cells are marked red (scale bar: 50 μm). **c** False-color representation of the *z*-position of every pixel in the minimum projection. This information is used to estimate the *z*-position of motile NK cells (cell outlines are indicated in black). **d** Time-lapse image sequence of an NK cell-mediated killing of a K562 tumor cell, taken from one out of six similar, independent experiments. **e** Same as **d**, with colored cell morphologies as a guide for the eye.

**Fig. 3 Influence of 3-D NK cell motility on cytotoxicity. a** Motile fraction of fresh (blue) and cryopreserved (red) NK cells from $n = 10$ independent experiments from five subjects and five different expansions in 1.2 mg/ml collagen gel ($p = 0.002$; two-sided nonparametric Wilcoxon signed-rank test for paired data). Each symbol represents mean ± se from cells measured in five field of view, with ~80 cells (motile plus nonmotile) in each field of view. In total, $n = 1248$ fresh motile cells and $n = 122$ cryopreserved motile NK cells were measured. Paired (fresh vs. cryopreserved) data from each subject and expansion are connected by lines. **b** NK cell cytotoxicity (as measured in a chromium-release assay) as a function of the motile NK cell fraction for fresh (blue) and cryopreserved (red) NK cells. Different NK-to-target cell ratios are noted above each graph. Dashed lines indicate a power-law fit of the form $f(x) = a \cdot x^b$. $R^2$ values are computed in log–log space. Statistical significance assuming a power-law exponent of $b = 0$ as null hypothesis: 20:1: $p = 0.0013$; 10:1: $p = 0.0028$; 5:1: $p = 0.0013$; 2.5:1: $p = 0.00045$ by two-sided nonparametric Wilcoxon signed-rank test for paired data. Source data are provided as a Source data file.

from the number of live target cells at the beginning and the end of the experiment (see Eq. (2) in "Methods" section and Fig. 4c). Note that the killing rate of an individual NK cell cannot be determined in this assay as the cells are free to migrate into and out of the microscope's field of view during the observation period. Normalizing this exponential decay rate to 1 h, as well as to one motile NK cell and one living target cell per $10^6$ μm³ volume of tissue (corresponding to a cube of 100 μm length), we arrive at a measure of killing efficiency that is not biased by varying cell concentrations or fractions of motile cells (see Eq. (3) in "Methods" section and Supplementary Fig. 6). We find a 26% decrease of cytotoxicity in a 3-D environment after cryopreservation, but this difference is smaller than the difference detected in the classic cytotoxicity assays and is not statistically significant

(Fig. 4d). Therefore, our data demonstrate that NK cells that remain motile after cryopreservation retain their cytotoxic cell function. However, if we compute a composite killing efficiency that considers both motile and nonmotile NK cells, we find a significant decrease of effective cytotoxicity in a 3-D environment by a factor of 5.6 (Fig. 4e), which directly reflects the sixfold decrease in NK cell motility.

## Discussion
In this study, we report that ex vivo-expanded NK cells retain their ability to induce target cell death after cryopreservation (as measured in a degranulation assay), but suffer a significant decrease of their cytotoxic function (as measured in a chromium-

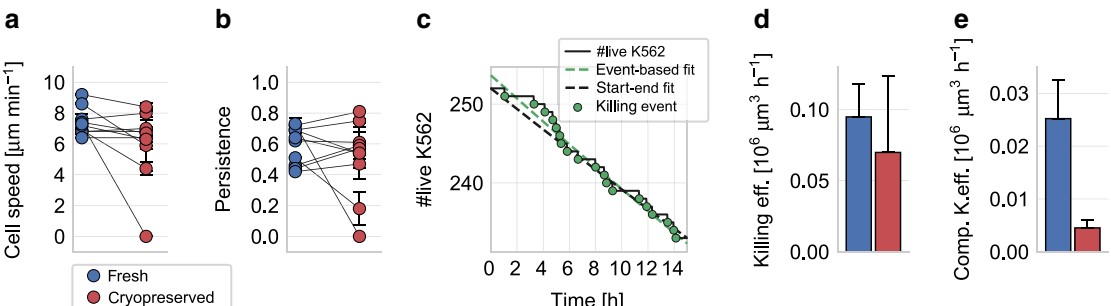

**Fig. 4 NK cell migration and cytotoxicity in 3-D collagen gels. a** Migration cell speed of fresh (blue) and cryopreserved (red) NK cells from $n = 10$ independent experiments from five subjects and five different expansions in a 3-D collagen gel. Each symbol represents mean ± se (measured in the $x$–$y$-imaging plane) from cells measured in five field of view, with ~80 cells (motile plus nonmotile) in each field of view. Migration cell speed and persistence is computed from on average $n = 125$ motile fresh cells and $n = 12$ motile cryopreserved cells for each subject and expansion (each data point). In total, $n = 1248$ motile fresh and $n = 122$ motile cryopreserved NK cells were measured. Paired (fresh vs. cryopreserved) data from each subject and expansion are connected by lines. **a** Cell speed ($p = 0.076$; two-sided nonparametric Wilcoxon signed-rank test for paired data). **b** Migration directional persistence ($p = 1.0$; two-sided nonparametric Wilcoxon signed-rank test for paired data). **c** Number of live K562 target cells (black line) in a representative experiment as a function of measurement time. Killing events are marked by green circles. The dashed green line indicates an exponential fit of the individual killing events, the black dashed line indicates an approximation using an exponential curve based only on the number of live targets at the beginning and the end of the measurement time. **d** Estimated killing efficiency (weighted mean ± se determined by bootstrapping) of fresh and cryopreserved motile NK cells against K562 target cells embedded in a 3-D collagen gel ($p = 0.56$; two-sided bootstrapping; $n = 6$ independent experiments from four subjects and three different expansions). **e** Estimated composite killing efficiency (weighted mean ± se determined by bootstrapping) of fresh and cryopreserved NK cells (considering both motile and nonmotile NK cells) against K562 target cells embedded in a 3-D collagen gel ($p = 0.025$; two-sided bootstrapping; $n = 6$). Note that the killing efficiency of the individual experiments is not shown in **d**, **e** as the reported mean values are weighted by the respective cell concentrations (see "Methods" section). Individual values are provided as a Source data file.

release assay). By complementing standard assays of NK cell function with measurements of NK cell motility and cytotoxicity in a 3-D environment, we are able to attribute this decrease of cytotoxic function to an impairment of NK cell motility. Specifically, we find a dramatic sixfold decrease in the fraction of motile NK cells after cryopreservation. We further show that the small remaining population of motile NK cells retains its cytotoxicity after cryopreservation.

NK cells are generally cryopreserved using 10–20% dimethyl sulfoxide (DMSO) solution as a cryoprotectant at a slow freezing rate of −1 °C/min (ref. [22]). Several groups have reported changes in cell viability and/or function in non-expanded (primary) NK cells after cryopreservation. Specifically, the NK cell compartment in PBMC loses cytotoxic function after thawing, while retaining cell viability[23,24]. Likewise, purified primary NK cells also show reduced cytotoxicity after cryopreservation, although cytotoxicity is significantly improved after an overnight recovery culture in the presence of IL-2 (refs. [16,25]). Treatment of patients with IL-2 after adoptive transfer of NK cells, however, did not translate into clinical success beyond the effect of IL-2 alone[7,10]. In addition, non-expanded NK cells may be susceptible to cryopreservation-induced phenotype changes, since cryopreservation induced a transient increase in CD56+ CD16− NK cells in peripheral blood from hematopoietic stem cell transplantation recipients[26]. This phenotypic change is similar to the CD16 decline we noted after freezing and thawing, which may be the net result of activation-induced cleavage of CD16 (ref. [19]). Thus far, attempts to optimize current NK cell cryopreservation protocols have had limited success[16,27].

Overall, the presented findings point to the cryopreservation step as a major cause for the current failure of NK cell therapy in patients with solid tumors. In the future, this problem may be solved by sterile expansion of NK cells in a closed bioreactor that circumvents the cryopreservation step altogether. Moreover, our data show that the fraction of motile NK cells largely determines the effective cytotoxicity, and is therefore a crucial parameter when deciding on the number of cells to be administered to the patient. Finally, this work demonstrates that time-lapse imaging

of NK cells and tumor cell killing events in a 3-D matrix may serve as a better predictor of NK cell function compared to standard cytotoxicity assays.

## Methods

**PBMC isolation and storage**. NK cells are generated from PBMCs from 11 healthy donors (Department of Transfusion Medicine, University Hospital Erlangen, Germany; IRB approval number 147_13B). We have obtained informed consent from all participants. Specifically, leukocyte reduction system chambers are obtained from the department of transfusion medicine, from which PBMCs are isolated. PBMCs are resuspended in human serum albumin (Baxter) at a concentration of $5–10 \times 10^6$ cells/ml. A total of 500 μl of cell suspension is filled into a 1 ml cryovial (Nalge, Nunc), and 500 μl of freezing medium (55.5 vol% human serum albumin, 25.0 vol% DMSO, 8% (w/v) glucose) is added. After gently mixing, the vials are transferred into a freeze container (Mr. Frosty, Thermo Scientific, which allow for a cooling rate of −1 °C/min) and stored at −80 °C. Cells are used within 65 days after cryopreservation.

**NK cell expansion**. After thawing the PBMC samples (which includes NK, NKT, and T cells), cells are expanded in the presence of irradiated K562-mbIL15-41BBL feeder cells (gift from Prof. D. Campana, Department of Pediatrics, University Hospital Singapore; formerly St. Jude Children's Research Hospital, Memphis, TN, USA) for 14 days in RPMI 1640 medium supplemented with 10% fetal bovine serum, 20 μg/ml gentamycin and 1% L-glutamine (hereafter called cRPMI), and 200 IU/ml human IL-2 cytokine[28]. K562 feeder cells are confirmed negative for mycoplasma contamination using the Venor GeM Classic detection kit (Minerva Biolabs). This expansion process is performed five times in the case of 1 donor, three times in the case of 3 donors, two times in the case of 3 donors, and one time in the case of the remaining 4 donors, resulting in a total of 24 donor/expansion combinations.

**NK92 cell line**. NK-92 cells (purchase from ATCC CRL-2407) are cultured for 3 weeks prior to measurements in Alpha-MEM medium (Stemcell Technologies) with 15% fetal calf serum, 15% horse serum, 500 IU/ml human IL2-cytokine, and 1% penicillin–streptomycin solution (10,000 units/ml penicillin, 10,000 μg/ml streptomcycin).

**Cryopreservation and thawing**. Expanded NK cell aliquots ($5–10 \times 10^6$ cells/ml) are either directly measured (in the following referred to as "fresh"), or are frozen, thawed, and then measured (in the following referred to as "cryopreserved"). We conduct paired experiments (fresh versus cryopreserved) for all assays, but due to technical difficulties in some of the assays, the number of paired experiments varied as noted in the figure legends. Freezing of expanded NK cells is performed as described above for PBMCs. On the following day, cryopreserved expanded NK

cells are rapidly thawed in a 37 °C water bath until a small visible ice chip remains, and then are dropwise transferred into 10 ml of cRPMI, which is preheated to 37 °C. Cells are centrifuged at $300 \times g$ for 5 min and resuspended in cRPMI.

**Flow cytometry**. Cell viability of fresh and cryopreserved NK cells is assessed by staining with the Zombie NIR dye (dilution 1:1000; Biolegend). Fresh and cryo-preserved NK cells are phenotypically characterized as described in refs. [28,29] by staining with directly conjugated mouse anti-human antibodies against CD3 (clone UCHT1; dilution 1:50; Biolegend), CD56 (clone HCD56; dilution 1:50; Biolegend), and CD16 (3G8; dilution 1:50; Biolegend). NK cells are defined as CD3− and CD56+ cells (Supplementary Fig. 7). A minimum of 10,000 cells are analyzed using a BD Canto II flow cytometer (BD Biosciences) and Flowjo Software (FLOWJO, LLC Data analysis software).

**CD107a degranulation assay**. A total of $1 \times 10^6$ expanded NK cells are incubated for 6 h at 37 °C, 5% $CO_2$, 95% RH with cells from the myeloid cell line K562 (gift from Dr. J.J. Bosch, Department of Medicine 5, University Hospital Erlangen) at an NK-to-K562 cell ratio of 20:1 and 5:1 in a final volume of 500 µl cRPMI supple-mented with anti-CD107a antibody (clone H4A3, 10 µl/ml, BD Biosciences). K562 cells are confirmed negative for mycoplasma contamination. To prevent protein secretion and degradation of internalized CD107a, monensin (1 µM) and brefeldin A (10 ng/ml, both from Sigma) are added after 1 h of incubation. NK cells alone serve as a negative control, and NK cells stimulated for 6 h with phorbol 12-myristate 13-acetate (PMA, 50 ng/ml) and ionomycine (250 ng/ml, both from Sigma) serve as a positive control for anti-CD107a antibody binding. After 6 h of incubation, cells are harvested, washed, resuspended in 50 µl PBS, and stained with live–dead Zombie NIR (BioLegend), anti-CD56 (clone CHD56, BioLegend), and CD16 antibody (clone 3G8, BioLegend). Samples are analyzed using a Becton Dickinson FACS CANTOII flow cytometer and Flowjo software.

**Chromium-release assay**. K562 cells are labeled with radioactive (150 µCi, 5.55 MBq) sodium chromate (20 µl/condition, 5 mCi/ml, Perkin Elmer) for 1 h. After incubation, cells are washed two times and incubated for an additional 30 min to reduce spontaneous chromium release. Labeled cells are then plated at a density of 5000 cells/well in 100 µl cRPMI in a 96-well U-bottom plate. Fresh expanded or cryopreserved NK cells are added at NK-to-target cell ratios of 20:1, 10:1, 5:1, and 2.5:1 to give a final volume of 200 µl per well. After 0.5, 1, 2, 3, or 4 h of incubation, 100 µl supernatant is mixed with 100 µl scintillation cocktail (Perkin Elmer) in a 96-well sample plate (Perkin Elmer). Release of radioactive chromium-51 is measured using a gamma-counter (Perkin Elmer), and the fraction of lysed target cells is calculated as the ratio of (experimental release − spontaneous release)/(maximum release − spontaneous release). Spontaneous release is measured from 5000 labeled K562 cells without addition of NK cells, and maximum release is measured from 5000 labeled K562 cells that are lysed with 100 µl 1% Nonidet P-40 (Sigma). All experiments are performed in triplicates.

**3-D cell motility assay**. We suspend 150,000 fresh or cryopreserved NK cells in 2.5 ml of a 1.2 mg/ml collagen solution or in 2.5 ml of 9 mg/ml carbomer hydrogel (Ashland 980 Carbomer, Covington, USA) in each well of a tissue-culture treated six-well plate (Corning). The collagen solution is prepared from a 2:1 mixture of rat tail collagen (Collagen R, 2 mg/ml, Matrix Bioscience) and bovine skin collagen (Collagen G, 4 mg/ml, Matrix Bioscience). We add 10% (vol/vol) sodium bicar-bonate (23 mg/ml) and 10% (vol/vol) 10× RPMI (Gibco). For a final collagen concentration of 1.2 mg/ml, we dilute the solution before polymerization with a mixture of one volume part NaHCO₃, one part 10× cRPMI, and eight parts H₂O (ref. [30]) and adjust the solution to pH 9 with NaOH. After polymerization at 37 °C, 5% $CO_2$, and 95% RH for 1 h, 1.5 ml of RPMI medium (for primary NK cells) or 1.5 ml of Alpha-MEM medium (for NK92 cells) is added to each well of a six-well plate.

Carbomer hydrogel is prepared by mixing carbomer powder with RPMI 1640 medium (9 mg/ml). The pH is titrated to a value of 7.4 with 10 M NaOH. After mixing-in the cells, the migration assay is started without a waiting period. In the case of cells embedded in collagen, we add a waiting period of 3 h to ensure that the NK cells have adapted to the collagen gel, attained their characteristic elongated shape and recovered their full migratory potential (Supplementary Fig. 8). For the migration assay, the six-well plate is transferred to a motorized microscope (Applied Scientific Instrumentation, USA, equipped with a 10× 0.3 NA objective (Olympus) and an Infinity III CCD camera, Lumenera) that is placed inside an incubator, and time-lapse imaging is started. We perform z-scans (10 µm apart) through the 1 mm thick gel every 30 s for a duration of 5 min. Afterward, another randomly chosen position is selected, and time-lapse imaging continues. In total, six positions are imaged in each well. Fast image acquisition of z-scans is achieved by taking images at 27.5 frames per second, while the focal plane of the microscope is moving with a constant speed through the gel along the z-axis. We do not store the entire z-stack of images, but instead store the lowest intensity value of each pixel along the z-direction (minimum intensity projection, Fig. 2a, b), and the z-position where the lowest intensity value is found (Fig. 2c). The z-position information aids in the tracking of two individual cells when their paths seem to

cross in the minimum intensity projection images. In a similar way, we compute the maximum intensity projection.

For analysis, we detect individual NK cells using a convolutional neural network that is trained on 70 manually labeled minimum/maximum intensity projections with ~100 cells in each projection. We assume a minimum diameter of nonmotile NK cells of 4 µm to separate them from smaller cell fragments (Supplementary Fig. 9 and Supplementary Movie 3). In addition, we perform data augmentation by image flipping and zooming. The network is based on the U-Net architecture[31]. The labeling accuracy of the network is 94% (F1-score), which is comparable to the manual labeling accuracy of the investigators involved in this study ($n = 4$ subjects). We then connect the x,y positions of detected NK cells between subsequent images to obtain migration trajectories (Fig. 2b and Supplementary Movie 2). An NK cell is classified as motile if it moves away from its starting point by ≥13 µm within 5 min. The cell speed is determined as the diagonal of the bounding box of each cell trajectory, divided by the measurement time of 5 min. Directional persistence is determined as the average cosine of the turning angles between consecutive cell movements. Zero persistence corresponds to random motion, whereas a persistence of unity corresponds to ballistic motion.

As the expanded cell populations contain small fractions of other cell types (as measured with flow cytometry, Fig. 1b), we perform confocal microscopy of an expanded cell population that is stained using CD56-APC and CD3-Alexa488 antibodies, and is embedded in a 3-D collagen gel. We find that 82% of all motile cells are NK cells (Supplementary Fig. 10).

**3-D cytotoxicity assay**. We mix 600,000 fresh or cryopreserved NK cells together with 120,000 K562 cells in 2.5 ml of collagen (1.2 mg/ml), as described for the 3-D motility assay. z-stack time-lapse imaging is performed at four positions in each well in parallel, at a rate of two frames per min for 15 h in total.

To automatically detect the number of motile NK cells, as well as the number of live and dead K562 cells in each frame, we train a convolutional neural network with 48 minimum and maximum intensity projection images, in which all motile and nonmotile NK cells, as well as live and dead K562 cells are manually labeled. Living K562 cells are characterized by a round appearance and small (1–2 µm) wiggling motion. When a K562 cell, after being contacted by an NK cell, starts to bleb, change shape, shrink, expand, stop wiggling, or change its bright-field contrast due to refractive index changes, we consider the cell as dead or as undergoing lysis or apoptosis (Fig. 2d, e and Supplementary Movie 3). We complement this manually labeled data set with 11 images in which K562 cells are fluorescently labeled using Hoechst 33342 dye (incubation with 2.5 µg/ml for 20 min followed by two washing steps in PBS), 11 images in which NK cells are fluorescently labeled using Hoechst 33342 dye, and one image of dead K562 cells in which cell death is induced by adding 0.01% TritonX. The complete data set is used to train a convolutional neural network based on the U-Net architecture[31]. As nonmotile NK cells are smaller, but otherwise appear similar to K562 cells, we only consider K562 cells that are larger than 14 µm in diameter for the analysis (Supplementary Fig. 9). An example demonstrating the performance of the automated detection and labeling is shown in Supplementary Movie 3. For evaluating NK cytotoxicity, each NK cell-mediated killing event identified by the network is visually inspected and confirmed, using the image-annotation software ClickPoints[32].

To evaluate the accuracy of determining target cell death in bright field images, we add the dye NucRed Dead 647 (Ready Probes, Thermo Fisher) to the media immediately after the polymerization of the collagen gel and also at the end of the measurement (Supplementary Fig. 11). At the beginning and at the end of the 15 h measurement period, an image stack is recorded. All K562 cells that are classified as living based on the bright-field criteria are stained negative. Therefore, the false-negative error rate is 0% (none of the dead cells are labeled falsely as living). However, from the K562 cells which are classified as dead based on the bright-field criteria listed above, 10.6% are stained negative, possibly because they still have an intact cell membrane despite undergoing apoptosis. Therefore, we potentially overestimate the killing rate (see below) by up to 10%.

We describe the decline in the concentration of live K562 cells due to NK cell-mediated killing during the 15 h observation time as a first-order enzyme reaction of the form $[NK] + [K562_{live}] \xrightarrow{k} [NK] + [K562_{dead}]$, whereby the NK cells serve as the enzyme catalyzing this reaction, and $k$ is the reaction rate (the killing efficiency). We assume that $k$ remains constant and that the cytotoxicity of NK cells does not exhaust over time, which is an oversimplification, but justified by the high ratio of NK cells to target cells. Moreover, we consider the average concentration of motile NK cells ($[NK]$) as being constant throughout the measurement. Finally, we assume that the reverse reaction does not take place, implying that K562 cells do not recover once they are undergoing apoptosis or lysis. Therefore, if a K562 cell forms small temporary blebs that disappear after a while, so that the cell appears alive for the remainder of the experiment, this cell is counted as living.

The dynamics of the above killing reaction can be described with a first-order differential equation, which has the solution

$$k = \frac{\ln[K562_{live,t=0}] - \ln[K562_{live,t=T}]}{[NK_{motile}] \cdot T}. \tag{1}$$

The only free parameter is the killing efficiency $k$ that we compute from the concentration (# of cells per $(100 \, \mu m)^3$ of gel volume) of live K562 cells at the

beginning ($t = 0$) and the end of the experiment ($t = T$), and the average concentration of motile NK cells.

As individual experiments $i$ have different cell concentrations, we compute weighted mean killing efficiencies

$$k_{\text{avg}} = \frac{\sum_i k_i \cdot [\text{K562}_{\text{live},t=0}]_i \cdot [\text{NK}_{\text{motile}}]_i}{\sum_i [\text{K562}_{\text{live},t=0}]_i \cdot [\text{NK}_{\text{motile}}]_i}, \tag{2}$$

for both fresh and cryopreserved NK cells. Below, we also report an apparent or composite killing efficiency for all NK cells regardless of their ability to move, which is computed according to

$$k_{\text{composite}} = \frac{\ln[\text{K562}_{\text{live},t=0}] - \ln[\text{K562}_{\text{live},t=T}]}{([\text{NK}_{\text{motile}}] + [\text{NK}_{\text{nonmotile}}]) \cdot T}. \tag{3}$$

Since only motile NK cells can reach their target cell to kill, this composite killing rate combines cytotoxicity and motility in one parameter.

**Statistical tests**. Unless noted otherwise, all statistical tests use the two-sided nonparametric Wilcoxon signed-rank test for paired data. The correlation between NK-to-target ratio and cytotoxicity reported in Fig. 1d is determined by Spearman's rank-order correlation; the corresponding test statistic is based on a $t$-distribution to account for the small sample size and to avoid over-rejection. Significant differences for the killing efficiency reported in Fig. 4d, e are determined by bootstrapping (two-sided test). Differences are considered as statistically significant for $p < 0.05$.

**Reporting summary**. Further information on research design is available in the Nature Research Reporting Summary linked to this article.

## Data availability

Data supporting the findings of this study are available within the article and its Supplementary information files or from the corresponding author upon reasonable request. Source data are provided with this paper.

## Code availability

The software code for data analysis is available from the corresponding author on request.

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

## Acknowledgements

This work was supported by the National Institutes of Health grant HL120839, the DFG grant ME 1260/11-1, the DFG Research Training Group 1962 ("Dynamic interactions at biological membranes: from single molecules to tissue") and the interdisciplinary center for clinical research (IZKF; project number J37). We thank Jacobus J. Bosch and Dario Campana for providing the tumor cell lines, and Torsten Tonn for advice on NK92 cell experiments.

## Author contributions

C.J.V., B.F., and G.S. designed the study. S.R. performed cell expansion, cryopreservation, cytometry, degranulation, and chromium-release assays. C.M., B.F., T.C., A.M., F.H., and N.B. developed, and performed the 3-D motility and cytotoxicity assays. L.H., A.W., S.S., and S.R. developed the neural network-based analysis method. T.E.A. developed the

migration assay for carbomer hydrogels. C.M., T.C., L.H., and S.S. performed data analysis. C.M. and T.C. generated the figures. C.M., T.C., B.F., and C.J.V. wrote the manuscript.

## Funding

## Competing interests

The authors declare no competing interests.
