## [Peer Review File · Nature Communications]

Reviewers' comments:

Reviewer #1, expert in NK cell therapies (Remarks to the Author):

As I understand, the manuscript focuses on 10 NK cell expansions (some thrice from the same donor, some twice, some single expansion) with a K562 feeder line expressing mbIL-15 and 41BBL with antibiotics and 200 IU IL-2.

The results look interesting. The interpretation is overreaching and that is probably the biggest issue with the paper. The authors need to tone down their 1-size-fits-all conclusion to this specific protocol that they utilize, or, preferentially, test and confirm with other expansion protocols with various conditions to be able to make these conclusions. Also, testing directly after thawing is not the optimal mimicry. The cells get in circulation, either extravasate or go through lung capture/escape before ending up in the tumor site in a standard IV delivery. The experimental design should mimic that.

Reviewer #2, expert on NK cell biology (Remarks to the Author):

In this manuscript the authors describe that impaired cytotoxicity of cryopreserved human natural killer cells is as a result of decreased cell migration. Cell migration assays in 3D are combined with direct visualization of target cell killing and statistical analysis of the correlation between motility and cytotoxicity. The finding that cryopreservation affects NK cell function and migration is an important one for immunotherapy, as the authors appropriately point out. The manuscript is well-written and the approaches that are taken include novel imaging and analyses of human NK cells in 3D that are rarely undertaken. The authors further show that modeling cell migration in relation to target cell viability uncovers a relationship between retention of migration and cytotoxic function. Despite these strengths, there are some weaknesses that detract significantly from the findings of the paper. In particular, the significance of impaired migration in a 3D collagen gel and how this relates to target cell killing in conventional Cr51 assays, which are performed in a round-bottomed plate with non-adherent lymphoid targets, is not demonstrated. Finally, the data would be strengthened by the inclusion of more robust numbers of measurements, as it seems that many of the main conclusions are drawn from as few as 10 cells from a single donor.

Major comments

1. It is unclear that the relationship between migration in a 3D collagen gels translates to cytotoxicity in a Cr51 assay in a U-bottomed tissue culture plate, as these environments have very different mechanical and biochemical properties. It would strengthen the study significantly to show this link. Similarly, the authors show that decreasing the effector to target ratio leads to decreased cytotoxicity by the cryopreserved cells only, whereas fresh cells are able to mediate killing at these ratios. By this rationale, assays with fresh cells performed at shorter timepoints, where serial killing and migration are presumably not as prevalent, should similarly equalize the killing capacity of cryopreserved cells. This would better define whether decreased lytic capacity is correlated with decreased migration or a direct function of it.

2. The finding that the fraction of cells that remain motile have intact cytotoxicity is not directly shown, but instead inferred from measurements of target cell death and cell migration. Therefore, the statement that "the small population of motile NK cells following cryopreservation retains cytotoxicity" (Discussion, line 144) has not been directly demonstrated.

3. The authors cite in the introduction that "cryopreservation has been shown to have no significant effect on NK cell cytotoxicity measured with classical degranulation and chromium-release assays" (Line 54). The citation for this statement refers to a paper testing the effects of cryopreservation on the function of in vitro derived cells, which have undergone significant

cytokine activation and may be functionally unique from freshly isolated peripheral blood NK cells. Other studies have cited effects of cryopreservation on fresh NK cell function and should be cited. There are also a number of studies that have reported the loss or retention of NK cell cytotoxicity under various conditions of cryopreservation. It would be appropriate to include a more comprehensive description of these in the Discussion.

4. Data is shown in bar graphs, but scatter or violin plots would give a more accurate depiction of the data, although would also highlight disparities in numbers of cells analyzed in some experiments (Figure 4A = 1248 fresh and 122 cryopreserved NK cells; is this correct?). Further, calculations of the frequencies of motile cells are based on as few as 10 measurements, so the 30% vs 5% calculations described are calculated from as few as 3 or 1 cells within a dataset of 10 (Fig. 3). Whether these data reflect representative or combined biological repeats is not described. While it's appreciated that the workflow described here is not high throughput, multiple donors/biological repeats and technical repeats should be included.

Minor comments

1. Figure 1 is not called in the paper.

2. The authors cite that CD16 is down-regulated and postulate this reflects decreased activation, however this CD16 down-regulation could reflect increased activation and subsequent cleavage of CD16 by matrix metalloproteases (Grzywacz, B., N. Kataria, and M. R. Verneris. 2007. CD56(dim)CD16(+) NK cells downregulate CD16 following target cell induced activation of matrix metalloproteinases. *Leukemia* 21: 356–359)

We thank the reviewers for their helpful comments and suggestions. Below, we give a point-by-point reply.

Reviewer #1, expert in NK cell therapies (Remarks to the Author):

The interpretation is overreaching and that is probably the biggest issue with the paper. The authors need to tone down their 1-size-fits-all conclusion to this specific protocol that they utilize, or, preferentially, test and confirm with other expansion protocols with various conditions to be able to make these conclusions.

We have followed the reviewer's suggestion and tested for the possibility that our findings (of reduced motile NK cell fraction after cryopreservation) may be an effect that is associated with our particular expansion protocol. To do so, we have repeated the same measurements with the NK92 cell line (purchased from ATCC). These cells are also cultured in the presence of IL-2 but at a higher concentration compared to the expansion protocol for primary cells (500 IU/ml as opposed to 200 IU/ml). The expansion medium for primary NK cells versus the culture medium for the NK92 cell line was as follows.

primary NK cells	NK92 cell line
RPMI 1640 medium	Alpha MEM without nucleosides
10% FBS	15% FCS and 15% HS
200 IU/ml IL2	500 IU/ml IL2
1% L-Glutamine	
20 µg/ml Gentamycin	
	1% Pen Strep Solution

Similar to our data with expanded primary NK cells, we find a significant decrease in the motile fraction of NK92 cells after cryopreservation, from 31% to 6% (Supplementary Fig. S4). This suggests that the reduced motile fraction after cryopreservation is not a unique effect associated with our specific expansion protocol.

To take the reviewer's critique one step further, we also agree that we cannot generalize our conclusions for other cryopreservation protocols. We followed a good manufacturing practice (GMP) conform protocol, which was developed and approved for the cryopreservation of T cells intended for clinical use (Wiesinger et al. *Frontiers Immunol* 2017; Wiesinger et al. *Cancers* 2019) but this of course does not rule out that other cryopreservation protocols may overcome the cell motility impairment. However, several groups have noted reduced NK cell cytotoxicity against K562 tumor targets after thawing, although no differences were found in NK cell viability before freezing and after thawing (Szmania S et al. *J Immunother.* 2015; Berg M et al. *Cytotherapy* 2009). For additional clarity, we added an explanatory and cautionary note to Introduction and extended the Discussion to highlight the need to either optimize current cryopreservation protocols or to circumvent the cryopreservation step altogether, e.g. by sterile expansion in a closed bioreactor.

Also, testing directly after thawing is not the optimal mimicry. The cells get in circulation, either extravasate or go through lung capture/escape before ending up in the tumor site in a standard IV delivery. The experimental design should mimic that.

We agree, and to address this point (i.e. to answer the question if cryopreserved cells regain their normal motility after some time after IV delivery), we have measured the collagen 3-D cell motility of NK cells from 3 subjects after different incubation times (6h, 24h and 48h after thawing) (Supplementary Fig. S3). We see a recovery in the motile fraction of cryopreserved

NK cells from approximately 2% (after 6h) to 8% (after 24h) and 7% (after 48h). By contrast, the motile fraction of fresh cells decreases from 36% (after 6h) to 29% (after 24h) to 20% (after 48h). Nonetheless, the fraction of motile cryopreserved NK cells at 48h after thawing is still nearly 3-times lower than the fraction of motile fresh NK cells, so our main conclusion still holds. Note that to mimic the situation in the human body, the cells are not stimulated with IL-2 during the incubation time, which may explain the overall decrease in the motile fraction of fresh cells from 36% to 20% over a time course of 48 h.

Reviewer #2, expert on NK cell biology (Remarks to the Author):

Despite these strengths, there are some weaknesses that detract significantly from the findings of the paper. In particular, the significance of impaired migration in a 3D collagen gel and how this relates to target cell killing in conventional Cr51 assays, which are performed in a round-bottomed plate with non-adherent lymphoid targets, is not demonstrated.

Regarding the significance of impaired migration in a 3D collagen gel: In our opinion, the clinical significance of the impaired 3-D migration in collagen is that it implies that the cryopreserved NK cells are largely unable to target solid tumors or any other target cells outside the blood stream. We realize from the reviewer's comment that we have failed to convey this information and have therefore sharpened our discussion.

Regarding the question of how impaired migration in a 3D collagen gel relates to target cell killing in conventional Cr51 assays, please see also our detailed answer to Major Comment #1 below. The Cr51 assay shows only a moderate reduction in the killing efficiency after cryopreservation, because the NK cells do not need to migrate substantial distances to reach target cells. This finding together with our degranulation assay data confirm that cryopreserved NK cells do not lose their cytotoxicity if they are in direct contact with a target cell (which is why cell therapy with cryopreserved NK cells is effective to treat leukemia and other blood-born tumors).

However, also in the CR51 cytotoxicity assay, some degree of migration may still be beneficial as the cells are not freely suspended but are crowded in a pellet. Indeed, we could show a significant correlation between Cr51 killing efficiency and 3-D motility in collagen for NK cells from different subjects and isolations (Fig. 3b). This means that NK cells from samples with higher 3-D motility in collagen also showed a higher Cr51 killing efficiency. We have furthermore performed additional experiments in a different hydrogel (carbomer) that mimics the situation in a CR51 assay cell pellet, and we have assessed the CR51 cytotoxicity at earlier time points as suggested by the reviewer – see below. Together, these data demonstrate the significance of impaired migration in a 3D collagen gel and how this relates to target cell killing in conventional Cr51 assays.

Finally, the data would be strengthened by the inclusion of more robust numbers of measurements, as it seems that many of the main conclusions are drawn from as few as 10 cells from a single donor.

For each expansion from each donor, we measured typically 500 cells (250 fresh and 250 cryopreserved cells), which gives us excellent statistical power. The reviewer is correct that a motile fraction of typically 5% for the cryopreserved fraction translates to only 12-13 motile cells out of these 250 cells. But even if we had discovered that no cell is motile after cryopreservation, we can base our conclusion (of an impaired motile fraction) on data drawn from 250 cells and not zero cells. Moreover, we used paired tests where we compare the cryopreserved cells with fresh cells, which for a typical motile fraction of 30% translates to 75 cells. This means that when we compare the migration behavior of the motile fraction of the cells (e.g. regarding speed, persistence), we have a data base of about $75+12=87$ cells. The reviewer is of course right that a low motile fraction results in larger standard errors for speed

and persistence, and with more data, small differences in speed and persistence between motile fresh and motile cryopreserved cells may reach statistical significance.

To include a more robust numbers of measurements, we performed experiments on 3 additional cell samples from different patients and isolations that more than doubled the number of measured cells. These measurements confirm our previous findings of an impaired motile fraction after cryopreservation. The data are shown separately in Supplementary Figs. S3 and S5 but are not included in the data set shown in Fig. 3,4 as we have not performed 3-D killing assays in collagen gels with those cells.

Major comments

1. It is unclear that the relationship between migration in a 3D collagen gels translates to cytotoxicity in a Cr51 assay in a U-bottomed tissue culture plate, as these environments have very different mechanical and biochemical properties. It would strengthen the study significantly to show this link.

To address the reviewer's question of how NK cell migration is affected by different 3-D environments with different mechanical and biochemical composition, we performed additional experiments with fresh and cryopreserved cells that are suspended in carbomer (a hydrogel-forming polymer based on acrylic acid) instead of collagen. Carbomer forms ~10 μm colloidal particles that are jammed to a viscoelastic (yield stress 10 Pa) hydrogel that is reminiscent of a cell pellet. Carbomer – in contrast to collagen – does not support the adhesion of cells to the matrix but only allows the cells to migrate in an amoeboid fashion. We find that the migration behavior of NK cells is similar between a 3-D collagen network versus a cell pellet-like colloidal visco-elastic hydrogel (Fig. S15). The motile fraction and migration persistence is slightly reduced in carbomer compared to collagen, but the effect of cryopreservation is similarly pronounced in carbomer as it is in collagen. Together, these data demonstrate that the reduced cytotoxicity in a Cr51 assay after cryopreservation is to a substantial degree attributable to a reduced motile fraction.

In addition, the data in Fig. 3B show a correlation between 3-D migration and Cr51 killing efficiency. Together with the data from the degranulation assay (which shows no significant impairment of cytotoxicity after cryopreservation), we interpret this correlation as being caused by those NK cells that are able to kill multiple target cells in a cell pellet beyond their immediate neighbors, for which some degree of motility may be of advantage. Hence, the Cr51 assay is a read-out of both, cytotoxicity and motility, and it is difficult to disentangle those effects.

By contrast, cytotoxicity and motility can be cleanly disentangled by directly imaging both, cell migration and individual killing events. Our data demonstrate that the reason for the 5.6-fold reduced effective killing efficiency (Fig. 4e) is a 6-fold reduced motile fraction (Fig. 3a). The killing efficiency of the cryopreserved motile cells is not significantly reduced (Fig. 4d). Taken together with the data from the degranulation assay (Fig. 2c), the ~2-fold reduced cytotoxicity seen in the Cr51 assay is predominantly the effect of a reduced 3-D motility and not so much due to a reduced "direct contact" cytotoxicity. We have included this argument in the revised manuscript.

Similarly, the authors show that decreasing the effector to target ratio leads to decreased cytotoxicity by the cryopreserved cells only, whereas fresh cells are able to mediate killing at these ratios. By this rationale, assays with fresh cells performed at shorter timepoints, where serial killing and migration are presumably not as prevalent, should similarly equalize the killing capacity of cryopreserved cells. This would better define whether decreased lytic capacity is correlated with decreased migration or a direct function of it.

This is an excellent suggestion. We have performed the Cr51 release assay at earlier time points (with NK cells from 5 different subjects) and indeed find that this tends to equalize the killing capacity of the cryopreserved cells especially at a low effector to target ratios of 2.5:1,

just as the reviewer predicted (Supplementary Fig. S1). This finding supports the notion that the decreased lytic capacity after cryopreservation as seen in the Cr51 release assay is to a large extent a direct consequence of a reduced motile fraction.

2. The finding that the fraction of cells that remain motile have intact cytotoxicity is not directly shown, but instead inferred from measurements of target cell death and cell migration. Therefore, the statement that “the small population of motile NK cells following cryopreservation retains cytotoxicity” (Discussion, line 144) has not been directly demonstrated.

We compute the killing rate from the number of killed target cells and the number of motile NK cells for each field-of view separately, not from a mean number averaged over many fields of view or many samples. We have moreover confirmed that prior to each target cell death, that particular cell was attacked by a motile NK cell. Hence, each target cell death can be directly related to an attack of a motile NK cell.

3. The authors cite in the introduction that “cryopreservation has been shown to have no significant effect on NK cell cytotoxicity measured with classical degranulation and chromium-release assays” (Line 54). The citation for this statement refers to a paper testing the effects of cryopreservation on the function of in vitro derived cells, which have undergone significant cytokine activation and may be functionally unique from freshly isolated peripheral blood NK cells. Other studies have cited and should be cited. There are also a number of studies that have reported the loss or retention of NK cell cytotoxicity under various conditions of cryopreservation. It would be appropriate to include a more comprehensive description of these in the Discussion.

Thank you for this suggestion, which we have followed in the revised manuscript. Specifically, we now discuss findings from other groups that have reported decreased cytotoxicity also in non-expanded primary NK cells:

NK cells are generally cryopreserved using 10% - 20% dimethyl sulfoxide (DMSO) solution as a cryoprotectant at a slow freezing rate of -1 °C./min. (El Assal R et al. Adv Sci 2019) . Several groups have reported changes in cell viability and/or function in non-expanded (primary) NK cells after cryopreservation. Specifically, the NK cell compartment in PBMC loses cytotoxic function after thawing, while retaining cell viability (Dominguez E et al. Biochem Soc Trans 1997; Mata M et al. J Immunol Methods 2014). Likewise, purified primary NK cells also show reduced cytotoxicity after cryopreservation, although cytotoxicity is significantly improved after an overnight recovery culture in the presence of IL-2 (Voshol H et al. J Immunol Methods 1993; Miller J et al. Biol Blood Marrow Transplant 2014). Treatment of patients with IL-2 after adoptive transfer of NK cells, however, did not translate into clinical success beyond the effect of IL-2 alone (Rosenberg S. A. et al. J. Natl. Cancer 1993, Parkhurst M. R. et al. Clin. Cancer Res. 2011). In addition, non-expanded NK cells may be susceptible to cryopreservation-induced phenotype changes, since cryopreservation induced a transient increase in CD56+CD16- NK cells in peripheral blood from hematopoietic stem cell transplantation recipients (Lugthart G et al. Blood 2015). This phenotypic change is similar to the CD16-decline we noted after freezing and thawing, which may be the net result of activation-induced cleavage of CD16 (Grzywacz B et al. Leukemia 2007). Thus far, attempts to optimize current NK cell cryopreservation protocols have had limited success (Miller S et al. Biol Blood Marrow Transplant 2014; Pasley S et al. Immunology Letters 2017).

4. Data is shown in bar graphs, but scatter or violin plots would give a more accurate depiction of the data, although would also highlight disparities in numbers of cells analyzed in some experiments (Figure 4A = 1248 fresh and 122 cryopreserved NK cells; is this correct?).

We followed the reviewer's suggestion and provide scatter plots of the data. We confirm that Fig. 4a shows the data from 1248 motile fresh cells and 122 motile cryopreserved cells. As explained above, a similar number of (motile plus non-motile) fresh versus cryopreserved cells were investigated (~ 5000 cells in this data set, plus another ~10.000 cells in the data set shown in Supplementary Fig. 3 and 5). The disparities in numbers of cells (e.g. for computing the migration speed of the motile fraction in Fig. 4a) is a consequence of the fact that the motile fraction of cryopreserved NK cells is much lower.

Further, calculations of the frequencies of motile cells are based on as few as 10 measurements, so the 30% vs 5% calculations described are calculated from as few as 3 or 1 cells within a dataset of 10 (Fig. 3).

The n=10 refers to 10 independent cell expansions, not to the number of motile cells (which are altogether 1248 fresh and 122 cryopreserved NK cells). This is now clarified in the figure legend and also made more clear by replacing the bar plots with scatter plots.

Whether these data reflect representative or combined biological repeats is not described.

We replaced Figs. 3a and 4a,b with the paired (fresh vs. cryo) data from each individual expansion. In the revised manuscript, we now clarify that we perform paired statistical tests where the data for fresh vs. cryopreserved cells are compared between each donor and expansion. We include in Supplementary Fig. S2 where we label the data for each subject and each isolation.

While it's appreciated that the workflow described here is not high throughput, multiple donors/biological repeats and technical repeats should be included.

Thank you for this suggestion. In the revised manuscript, we show and label the data for each donor (1-5) and each expansion (a,b,c) (Suppl. Fig. S2).

Minor comments

1. Figure 1 is not called in the paper.

Thank you. In the revised manuscript, Fig. 1 is called in the introduction (Page 4).

2. The authors cite that CD16 is down-regulated and postulate this reflects decreased activation, however this CD16 down-regulation could reflect increased activation and subsequent cleavage of CD16 by matrix metalloproteinases (Grzywacz, B., N. Kataria, and M. R. Verneris. 2007. CD56(dim)CD16(+) NK cells downregulate CD16 following target cell induced activation of matrix metalloproteinases. *Leukemia* 21: 356–359)

Thank you, this is an important point, and we now cite and discuss this paper (in Results and Discussion):

In Results: First, cryopreservation may affect NK cell activation and induce subsequent cleavage of CD16 by the activation of matrix metalloproteinases (Grzywacz B et al. Leukemia 2007), as indicated by the reduced number of CD16+ cells.

In Discussion: This phenotypic change is similar to the CD16-decline we noted after freezing and thawing, which may be the net result of activation-induced cleavage of CD16 (Grzywacz B et al. Leukemia 2007).

REVIEWERS' COMMENTS:

Reviewer #2 (Remarks to the Author):

The authors have significantly revised their manuscript and addressed my concerns. I congratulate them on an interesting and important contribution to the literature.

Reviewer #2 (Remarks to the Author):

The authors have significantly revised their manuscript and addressed my concerns. I congratulate them on an interesting and important contribution to the literature.

We thank the reviewers for their helpful comments and suggestions.